# Influence of Method of Treatment of Mandibular Condylar Fractures on Range of Articular Path Measured by Cadiax Device

**DOI:** 10.3390/jcm13133706

**Published:** 2024-06-25

**Authors:** Damian Niedzielski, Iwona Niedzielska, Daria Wziątek-Kuczmik, Maciej Kamiński, Stefan Baron, Sławomir Grzegorczyn

**Affiliations:** 1Department of Cranio-Maxillofacial Surgery, Faculty of Medical Sciences, Medical University of Silesia, 40-027 Katowice, Polanddkuczmik@sum.edu.pl (D.W.-K.); 2Department of Temporomandibular Disorders, Medical University of Silesia in Katowice; Traugutta sq. 2, 41-800 Zabrze, Poland; sbaron@sum.edu.pl; 3Department of Biophysics, Faculty of Medical Sciences in Zabrze, Medical University of Silesia, 19 H. Jordan Str., 41-808 Zabrze, Poland; grzegorczyn@sum.edu.pl

**Keywords:** TMJ, Cadiax, mandibular condylar fractures

## Abstract

**Background/Objectives**: The aim of this study was to evaluate the function of the treated temporomandibular joint based on the analysis of the image of the articular path using the Cadiax device depending on the choice of treatment method for unilateral condylar fracture of the mandible. **Methods**: Sixty patients who were treated for condylar fractures of the mandible at the Maxil-lofacial Surgery Department in Katowice were qualified for the analysis of the range of movements of the mandibular heads using the Cadiax device. From the group of patients who suffered fractures of the mandible, including condylar processes, patients were finally qualified for the measurement of the articular path of the injured and healthy joint according to strict criteria. **Results**: The condylar examination was performed in 20 patients who had conservative condylar fracture treatment and 40 patients who underwent various surgeries in the course of a single condylar fracture. The control group consisted of 20 patients whose mean values for the articular pathway measured for both sides were 12.73 and 12.69 and fell within the standard developed for healthy joints tested with the Cadiax device. **Conclusions**: We have achieved an almost ideal treatment for condylar fractures. We are also beginning to notice the need for rehabilitation of patients after this type of surgery.

## 1. Introduction

Mandible fractures are the most common fractures that occur in the facial part of the skull. Among them, condylar fracture accounts for 20–72% of cases [1,2]. Depending on the socio-economic level of the society, the most common causes are traffic accidents, beatings, falls from a height, and sport [3]. Until recently, due to the lack of appropriate radiological equipment, software supporting it, specialized tools, and specific guidelines, fractured condylar processes were treated mainly with conservative methods [4,5]. Technological progress allowed for precise individual diagnostics and qualification for a specific treatment procedure. Procedures became safer thanks to neuronavigation, planning based on stereolithographic models, and new individually designed systems for fixation [6,7]. The following are eligible for conservative treatment [orthopedic, surgically supported orthopedic and functional methods]: unilateral fracture of the condylar process of the mandible, incomplete fracture, unilateral fracture of the condylar process of the mandible without displacement, unilateral fracture of the condylar process of the mandible with displacement of the mandibular branch up to 5 mm (changed to 2 mm), displacement of the proximal fracture up to 50 degrees (changed to 30 degrees), bilateral fracture of the heads of the mandible without displacement or with moderate displacement not affecting occlusion, and bilateral fracture of the mandibular condylar processes without displacement or with moderate displacement [8]. The orthopedic method assumes the use of intermaxillary immobilization, which can be obtained through dental splints, wire bonds, orthodontic brackets, or additional elements such as a chin sling. They require the use of the patient’s own teeth; hence, their condition (mobility, soft tissue level, inflammation) must be appropriate for this. A modification of orthopedic treatment using the same assumptions is surgically supported treatment, in which instead of using the patient’s teeth, screws are used—skeletal anchors—placed in maxilla and mandible between the patient’s teeth. There is also a third solution, which is a kind of connection between the standard splint and bone anchorages—a modified splint with screw holes through which it is attached to the bone [9]. It should be emphasized that there is a significant difference in the time of placing such immobilization in patients, which directly translates into the comfort and stress associated with the visit. For the patient, the time spent in immobilization seems to be important. Fitting standard splints takes an average of 100–110 min, while IMF screws are put in in 8–16 min [9,10]. The recommended immobilization time ranges from 1 to 6 weeks [3 weeks on average], with lower fracture, lower fracture displacement, and younger age of the patient arguing in favor of shortening this period. It is clinically important that the mandible is properly immobilized in relation to the maxilla. IMF screws bring many advantages compared to dental splints, including no periodontal traumatization, the possibility of usage in patients with numerous prosthetic restorations, shortening the time of the procedure, increasing the safety of the procedure by reducing the risk of the operator being pricked, or simplicity in applying and removing them [9,10,11].

Functional treatment, which is becoming more and more popular, is the last of the conservative methods to combine orthodontic methods with physiotherapeutic treatment. Surgical treatment of condylar fracture is mandatory in patients in a coma, with severe mental illness, mental retardation, or in the absence of patient cooperation and in the treatment of multi-site and multi-organ injuries. We also qualify patients with limited access to the upper respiratory tract and with multiple jaw fractures with the necessity to properly rebuild the height of the mandible branches. An absolute indication (radiological criterion) for surgical treatment is a dislocated unilateral or bilateral fracture of the condylar process, shortening of the mandibular branch by more than 5 mm (2 mm), rotation of the proximal fracture by more than 50 degrees (30 degrees), and dislocation of the temporomandibular joint. On the basis of the clinical examination, we consider that an important criterion of qualification for surgery is a patient with occlusion disorders that cannot be resolved conservatively or through physiotherapy. Another important criterion is the presence of a foreign body in the fracture line, penetration of the fracture into the central fossa of the skull, and fracture of the skull base with cerebrospinal fluid flow [12,13,14,15,16].

The most frequently used surgical accesses to the area of the fractured condylar process are submandibular, retromandibular, and preauricular [17]. The sequence of these accesses corresponds to the increasing level of fracture. The literature recommends about 15 different accesses, which can be divided into two main groups—deep access and transverse access, the boundaries of which are determined by the course of the facial nerve. Deep accesses can be divided into supra-auricular, auricular, subauricular, and intraoral, while penetrating accesses can be divided into high, medium and low depending on which branches of the facial nerve they pass between [18,19]. However, in clinical practice, the previously mentioned three accesses are sufficient for most cases. A generally accepted and long-used method of surgical treatment of a fractured condylar process is a stable osteosynthesis with reposition to obtain correct anatomical relationships and correct bite. The differences arise with regard to the shape of the plate or screw, mostly made of titanium, through personalized solutions that consider a different structure of the condylar process and for a given type of fracture. Among the materials used for stable osteosynthesis today, titanium alloys or resorbable materials predominate. Resorbable materials in case of a fracture of the mandibular condyle are used much less frequently, mainly due to their poor mechanical properties. From titanium alloys, the operator has at their disposal a number of standard templates and customized solutions [20,21,22,23,24].

The aim of the study was to evaluate the function of the treated temporomandibular joint based on the analysis of the image of the articular path using the Cadiax device depending on the choice of the treatment method for unilateral condylar fracture of the mandible in retrospective material.

## 2. Materials and Methods

Sixty patients who were treated for condylar fractures of the mandible at the Maxillofacial Surgery Department in Katowice in 2010–2014 were qualified for the analysis of the range of movements of the mandibular heads using the Cadiax device. From the group of patients who suffered fractures of the mandible, including condylar processes, patients were finally qualified for the measurement of the articular path of the injured and healthy joint according to strict criteria. Due to the strict criteria for the group inclusion of patients, the groups became small, which may become a statistical problem; however, we want to ensure that the statistics were performed by academic researchers.

Eligibility criteria for research groups:A single fracture of the condylar process, regardless of the dislocation and degree of head dislocation in the TMJ, and regardless of the treatment method used.No other injuries of the facial skeleton upon admission to the clinic and in the period from the end of treatment to examination with the Cadiax device.Permissible simultaneous fracture of the mandibular body treated in a similar way to the fractured condylar process with anatomical effect.No further trauma to the TMJ in the period between the completed treatment of the condylar process fracture and the examination of the articular path for the needs of the analysis.Adult patients without systemic diseases.Patients without TMJ disorders based on the history.Presence of teeth in the upper and lower arch for lateral support.No malocclusion.Consent for testing with the Cadiax device.

Patients who experienced a unilateral fracture of the condylar process were classified into two research groups depending on the treatment method used:Group I: 20 patients qualified for conservative treatment who underwent condylar fracture treatment with the use of an elastic fixation based on standard dental splints for a period of 3 weeks. The eligibility criteria for conservative treatment in the above-mentioned period were mainly fractures without significant fragment dislocation and without occlusal disorders.Group II: 40 patients qualified for surgical treatment of mandibular condylar fractures. The eligibility criteria for surgical treatment in the above-mentioned period were mainly fractures with dislocation of the fragments disrupting the occlusion.

Surgically treated patients were divided into subgroups according to the type of treatment used at that time:Twelve patients: after fracture of the condylar process without dislocation of the head in the joint—internal fixation (IF).Eight patients: after fracture with displacement and fragmentation of the head of the condylar process possible for fixation—external fixation (EF)Thirteen patients: after shattering the head of the mandible that did not qualify for fixation—fragments of the head were removed without cutting branches and without additional procedures [R1—removal without cutting branches]Seven patients: after shattering the head of the mandible that was not eligible for fixation, fragments of the head were removed, mandible branch cut, shaped, and the positioned to rebuild height of the branch [R2—removal with cutting and branch modeling]

The control group included 20 healthy volunteers.

The eligibility criteria for the control group were as follows:Adult patients without systemic diseases.Patients without a TMJ injury and without symptoms suggesting TMJ disorders, including arthropathies.No malocclusion.Full dental arches.Consent for testing with the Cadiax device.

Qualification to groups and subgroups was made on the basis of the analysis of the patient’s medical history, including radiological documentation, dental diagrams, and laboratory tests. Over the years, we have improved treatment techniques; however, to standardize the study group, we qualified for this study using the same criteria. Due to the fact that in this period CT was rarely performed in mandibular fractures, the type of fracture was assessed on the basis of PA/lateral skull images and pantomograms using the classification according to Berchr and Krywins [low subcondylar fracture, high subcondylar fracture, and mandibular head fracture]. Patients qualified according to the strict criteria for inclusion in the measurements of the articular path with the Cadiax device were called for a follow-up examination. The age, sex, type of treatment performed, and the time elapsed between the injury and the treatment were recorded. After the measurements were taken on the Cadiax device, the range of mandibular movement was assessed for the injured joint and the healthy joint or both healthy joints in the control group.

The condylar path was measured using the Cadiax Compact II (Computer Aided Daignosis Axiography System) device, performing extraoral registration of the movements of the condylar processes (condylography) in three planes. Testing with this device takes about 10 min. The device consists of an upper facebow equipped with recording plates and a facebow attached to the mandible on a standard spoon with a thick silicone mass. Touch screen sensors collect data for the analyzer, which is connected to the computer via USB. The upper facebow is mounted in the reference plane running from the center of the mandible head (axis) to the point of the lowest edge of the orbit (orbitals). Each movement takes place from the same starting position, the so-called reference position, which is a clinically established central relationship, with no tooth contact, within the limits of the rotation of the condylar processes. The software allows you to obtain graphs for individual mandibular movements on the computer screen. Testing with this device is recommended in the study of patients with occlusal defects before and after orthodontic treatment in order to determine the effects of treatment and its impact on the function of the temporomandibular joint and to compare the movements in the temporomandibular joint in patients before and after surgery. The standard was adopted as the manufacturer’s reference value of 11–16 mm for forward movements and 10 mm for lateral movements.

Statistical analysis was performed using the Statistica 13.1 program (TIBCO Software Inc. Stanford Research Park, Palo Alto, CA, USA). The normality of the distributions of quantitative variables was tested using the Shapiro–Wilk test. The obtained variables are presented as arithmetic means (means) ± Standard Deviation (SD). Differences between groups in terms of quantitative unrelated variables were compared using the Student’s *t*-test for variables with normal distributions, while in the case of non-normality of the distribution of one of the variables, the Mann–Whitney U test was used. Statistically significant differences between the variables were considered for *p* < 0.05. In turn, the Pearson’s linear correlation coefficient was used to study the statistical dependence between two quantitative features. The research was approved by the Bioethical Committee of the Medical University of Silesia No. KNW/0022/KB1/87/I/18.

## 3. Results

The condylar path examination was finally performed in 20 patients (including 12 women and 8 men) who had a condylar fracture conservative treatment and 40 patients (11 women and 29 men) who underwent various surgeries in the course of a single condylar fracture (surgical group). The control group consisted of 20 patients, including 12 women and 8 men, whose mean values of the articular pathway measured for both sides were 12.73 and 12.69 and fell within the standard developed for healthy joints tested with the Cadiax device. Decisions about conservative treatment were made more often in women (12 women) than in men (8 men). Among the surgical procedures, the most common was internal fixation (12 patients), in which the fractured condylar process was set in a typical manner and fixed in place (trauma without dislocation of the head in the TMJ) and removal of the fragmented head without additional procedures (13 patients). The mean age of patients undergoing conservative treatment (31 years) was comparable to the group of patients undergoing external fixation (after cutting a branch and extracting a displaced/fragmented head, anastomosis with screws and insertion in an anatomical position; 30 years). Older patients more often experienced condylar fractures with a fragmented head that was impossible for treatment and therefore had it removed with or without cutting the branches of the mandible (mean age 33–42 years).

The mean values of the condylar path on the side of the injury were always lower compared to the healthy side in the group of patients who underwent surgery for a condylar fracture. The lowest values of the condylar path were recorded in the case of the fixation of the head fragments of the condylar process after cutting the branches, outside the organism. (4.539 trauma side/9.164 healthy side). In the conservative-orthopedic group, the movement of the heads in the joints was almost comparable, with the slight impairment on the trauma side (11.01 on the healthy side and 10.07 on the traumatic side). All data are summarized in Table 1.

Internal Fixation (IF)—typical fixation of fractures with titanium plates after adjusting the condylar process in anatomical positions.External Fixation (EF)—vertical cut of mandible branch, removal of fractured head, extracorporeal fixation of its elements with screws, extracorporeal attachment of the head of the convex branch, and fixation to the mandible.Removal without cutting branches [R1].Removal with cutting and modeling the branch [R2]—after cutting the branch, head fragment removal, modeling, and transposition of the branch.

In the group of patients treated with conservative method, significant differences were found in the mean values of the condylar path on the side of the injury compared to the healthy joint at the level of *p* < 0.05. The mean value for pathways of the condylar value on the side of the injury was 10.07 and on the healthy side was 11.01 (Figure 1).

The mean values of the condylar path for the temporomandibular joint, which was injured in the course of the condylar fracture and underwent surgical treatment using various techniques, were lower compared to the healthy side and amounted to 6.15 on the traumatic side and 10.36 on the healthy side. Significant deviations in the results of the condylar path measurements made it necessary to divide the surgical group into subgroups depending on the surgical method (Figure 2 and Figure 3).

In the surgical group [12 patients] treated with the method of internal fixation [typical titanium plate fixation of a fractured condylar process without head dislocation by submandibular access], the difference in the mean values of the articular path for the joint that had suffered an injury in the past and the healthy joint was statistically significant at *p* < 0.05. In the case of the operation of the fractured condylar process using the extracorporeal anastomosis technique [after a vertical incision of the mandibular branch, extracorporeal fixation of the broken condylar head fragments was performed with screws and the intracorporeal fixation of the stabilized head and the cut branch to the remaining stable parts of the mandible], significant differences in mean values of the articular path were found (*p* = 0.00283) between the traumatic and healthy sides to the detriment of the traumatic side. In the surgical group in which 13 fragmented heads of the condylar process were removed without cutting the mandibular branch, the condylar path at the site of the removed head was significantly impaired compared to the healthy side [*p* = 0.000027]. The mean value for the articular path on the side of the injury was 4.711, while on the opposite side it was 10.940 (Figure 4). 

In the surgical group in which the head of the condylar process was removed after cutting the mandibular branch, the condylar path at the site of the removed head was significantly impaired compared to the healthy side [*p* = 0.0471]. The mean value for the articular path on the side of the injury was 5.451, while on the opposite side it was 2.129 (Figure 5).

### Comparison of Average Values of The Condylar Path in the Group of Patients Treated with Conservative and Surgical Method in Comparison to the Control Group

A condylar fracture treated conservatively significantly impairs the articular pathway in comparison to the right and left healthy joint of the control group. A Student’s *t*-test for independent variables compared the value of the range of the condyle pathway on the trauma side of the conservative group in relation to the control group and gave the result *p* = 0.0482 (Table 2, Figure 6).

Surgery for a unilaterally fractured condylar process significantly impairs the articular pathway as compared to its value in healthy patients who did not suffer any trauma and did not have any TMJ disorders (Table 3, Figure 7 and Figure 8). The Mann–Whitney U test for independent trials comparing the value of the condyle path range on the trauma side of the surgical group in relation to the control group was *p* < 0.001.

Figure 5 shows box graphs representing the differences between the average temperature of the tongue apex (TA) in healthy and patients.

## 4. Discussion

The treatment of condylar fractures is still controversial. In general, it requires open reposition and immobilization or closed functional reposition [25,26,27,28]. According to the Chrcanovic procedure developed in 2012, the first method included patients with shortening of the mandibular branch and dislocation of the fragments, and the second included fracture among children without dislocation [29]. Despite clearly defined guidelines, there are subtle discrepancies in the radiological qualification for open reposition and osteosynthesis based on the measurement of shortening or rotation of the proximal fractured condylar process. In the same book, there are two models estimated on CT qualifying the patient for surgery and one developed by the author [for adults—variant I—shortening the vertical dimension above 5 mm and the rotation of the proximal part above 50°; option II—shortening the vertical dimension by more than 2 mm and rotation of the proximal fracture by more than 30°; option III—shortening the vertical dimension above 1 mm and rotation of the proximal fracture above 10°]. Already in another design by Osmola, slightly different values are given—5 mm and 30°. Abdel narrows these criteria down to 2 mm and 10° [30].

The treatment methods used so far have also evolved from conservative methods to the preference in most cases for surgical methods modified with modern fixation techniques and modification of surgical accesses (including the use of a neurostimulator, borrowed from the removal of the parotid gland for greater surgical safety) [31]. The conservative method is also experiencing evolution. In 1981, Joerg-Elard Otten used two steel hooks to fix the fractured condylar processes of the mandible, which he fastened with two screws, with the first one under the apical foramen and the second one in the area of the mandibular symphysis. Between the hooks, he placed a flexible rubber hoist [32,33]. Arthus and Berardo modified the Otten method in 1989 by introducing special intermaxillary fixation screws, commonly known as IMF screws [34,35]. Bicortical implants (TIB), used widely nowadays, differ in length, shape of the screw head, the material from which they are made (steel or titanium), and the way of insertion into the bone. [36] TIBs seem to be irreplaceable in situations limiting the use of dental splints, i.e., in patients with advanced periodontal disease, numerous missing teeth, extensive prosthetic restorations (bridges), as well as in cases of using full dentures [3,35,37]. Standard splints not only impede the maintenance of proper oral hygiene and provoke periodontitis but at the same time pose a real threat to the doctor related to needle stick injuries or injuries with a wire ligature [38]. The number of complications described in the literature after the use of TIB ranges from 2 to 4% [39]. These include local inflammation, loss of implant stabilization, iatrogenic root damage, limited peri-implant osteitis, mental nerve damage, and paresthesia at the site of implant insertion [40,41,42]. There is no comparison in the study within conservative method between IMF and splint due to lack of them at that time in Poland. Noteworthy is the fact that it is recommended to use bicortical screws for rehabilitation of patients after fracture of the condylar processes. A straight/oblique elastic fixation is stretched over them in the form of elastics with lateral support using acrylic plugs located on the molars on the injured side [43].

Surgical treatment of the fractured condylar processes of the mandible has also evolved, and it has also involved the shift from the methods of drastic removal of the broken heads of the mandible to their fixation or replacement with standard or individual endoprostheses produced thanks to the CAD/CAM system [44]. Until recently, condelectomy was recommended in the case of the destruction of the mandibular head, not having anything good to offer, only modeling the head from an elongated part of the mandibular branch, which in most cases did not bring good results [45].

In this analysis, it was considered justified to conduct research on a group of patients who were not subjected only to various treatment procedures in the course of condylar fracture, so it was decided to state which procedure of the procedure itself is the most harmful to the joint function. Hence, the analysis includes procedures for the removal of the condylar heads, modeling them from the displaced branch of the mandible, and even the extracorporeal anastomosis of the heads, which was very controversial at the time. Today, it is undisputed that neither of these methods should be considered, but the assessment of the consequences of their relation to an accepted method of internal fixation and conservative methods was interesting. And so, the greatest disturbances in the articular pathway examined with the Cadiax device occurred in the same order as the above-mentioned methods. Comparable disorders occurred after internal fixation as well as after conservative treatment, but it should also be taken into account that these patients were not rehabilitated. We do not know if rehabilitation works better for an orthopedic or surgical patient. This is the sound of the future and further research. The control group did not differ much from the standards set by the manufacturer of the Cadiax device. Individual differences were also not observed.

In the arsenal of the many possible tools for instrumental analysis of disorders in the stomatognathic system after condylar fractures, Cadiax seemed to be the most appropriate. Until now, despite the implementation of various methods of treatment of mandibular condylar fractures and observation of complications, and a constant discussion on the advantage of one method over the other, no studies, apart from clinical observation, have been conducted that would prove the right choice of a given therapy from the point of view of correct mandibular abduction. It would be interesting to include other methods of instrumental analysis, including EMG or T-scan, which could raise the rank of the observations. Perhaps in the future, we will evaluate the success of such sophisticated techniques of the surgical procedures themselves. The fundamental principle of treating all fractures is adaptation and immobilization. With regard to the condylar process, the overriding principle is its adjustment and immobilization with the restoration of the correct position of the mandibular head [1]. The conservative method is reserved for cases where the condylar head is in the joint and closed reposition is possible in the case of a fracture with a slight displacement or where the fracture is not displaced. In other cases, open reposition and osteosynthesis are required. The head is often pulled out of the joint and is located in the soft tissues. Whether we fix it extracorporeally or intracorporeally for the functioning of the system, it was once believed that it should not matter because the mandibular head loses contact with all joint structures. However, studies of the mandibular movement path proved that extracorporeal fixation significantly impairs the functions of the TMJ to a greater extent than intracorporeal osteosynthesis, and hopefully no one is considering such an operation without an attempt to replace the head with an endoprosthesis.

Treatment of condylar fractures is one of the most difficult surgical procedures in craniofacial injuries, and one of the post-operative complications that is not infrequently noted is malocclusion. Biomechanical analysis has shown that these disorders are often caused by insufficient treatment of the fractures running through the thinnest structure of the condylar process—the cervix. Therefore, the priority of the recent years in the treatment of this type of fracture has been a rigid and stable fixation. About 30 types of plates were dedicated for this purpose, but only 4 passed the good provision tests. Resolute experiments carried out by Kozakiewicz on stereolithographic models with the use of 30 titanium plates of the 2.0 system with a different number of eyelets/screws and shapes allowed the simulation of the loads and the indication of which shape with a given number of screws withstood the greatest loads on the condylar neck. The best solution suggested was straight, double plates arranged in parallel, then A- and X-shaped plates with at least 3–4 screws on the condyle side and 4–6 screws on the branch side. In the literature, many simulations of loads have been carried out after the use of various systems, including a given thickness of plates and screws, assuming that a greater number of plates is better and it is advantageous to arrange them in parallel with a large number of screws [46,47,48]. Two plates gained a decisive advantage over one in this case, which was confirmed in FEM tests, and Meyer also indicated the need to arrange them according to the lines of bone deformation [49,50,51]. The above-mentioned deliberations of scientists prove that today we not only have no doubts about what method of treatment is the best in a given case, but we also consider how many screws to use and what shape of the plate to use in a given type of fracture. The subject of analysis clearly indicated that regardless of the choice of treatment method, the injury itself, and also each procedure, impairs the function of the SS.

This is a retrospective study; it is inferior to a prospective study, which we are aware of. The research is still ongoing, and we are continuing and improving it; however, before publishing a prospective study, we wanted to know the opinion of other researchers.

## 5. Conclusions

Every injury affects the function of the temporomandibular joint. Conservative treatment limits the articular path to a lesser extent than surgery. Condylectomy without reconstruction or its external fixation is the most harmful procedure for the function of the temporomandibular joint. Internal fixation impairs the articular path to a similar extent to that of conservative treatment.

## Figures and Tables

**Figure 1 jcm-13-03706-f001:**
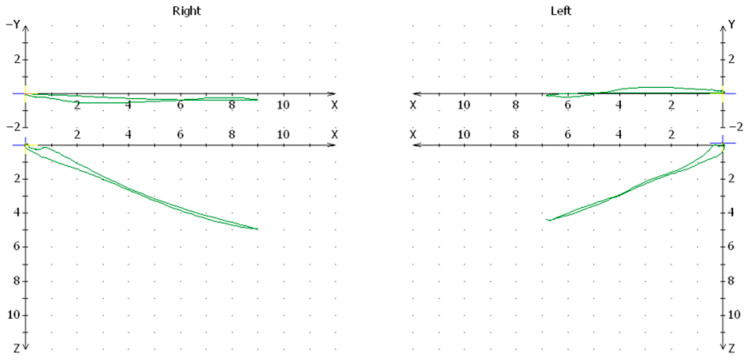
Cadiax articular path chart for one of the patients in the conservative-orthopedic treatment group.

**Figure 2 jcm-13-03706-f002:**
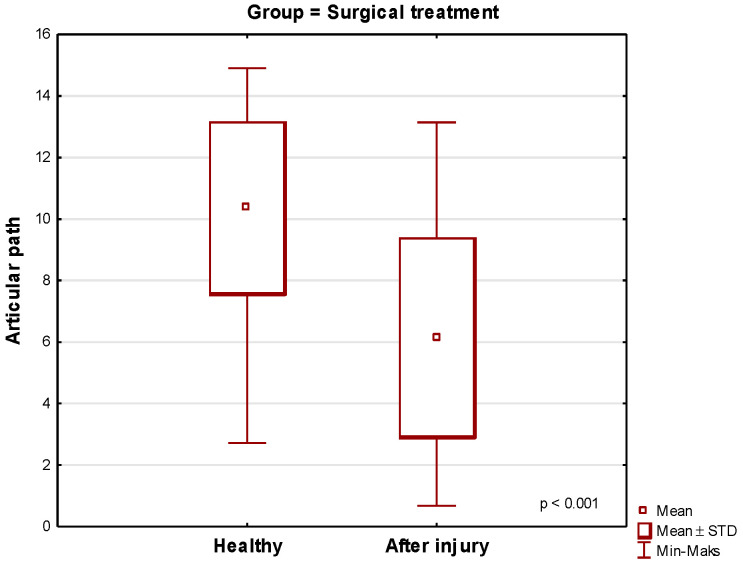
Mean values of the articular path measured after the surgical treatment of a single fracture of the condylar process for both sides of the body.

**Figure 3 jcm-13-03706-f003:**
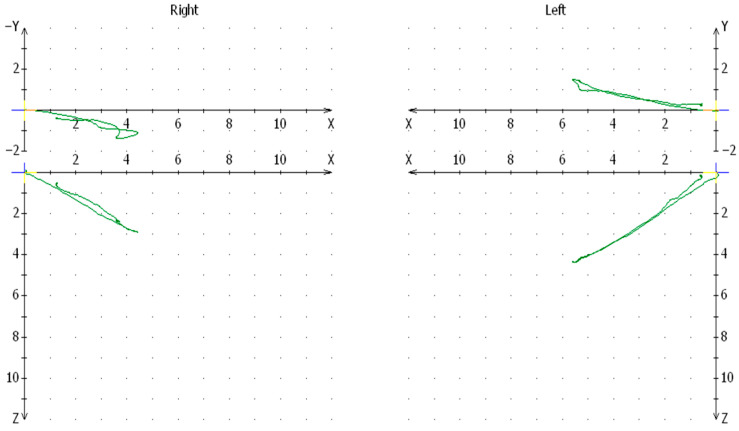
Diagram of the articular path in Cadiax for one of the patients from the group treated with the surgical method (right suction after trauma).

**Figure 4 jcm-13-03706-f004:**
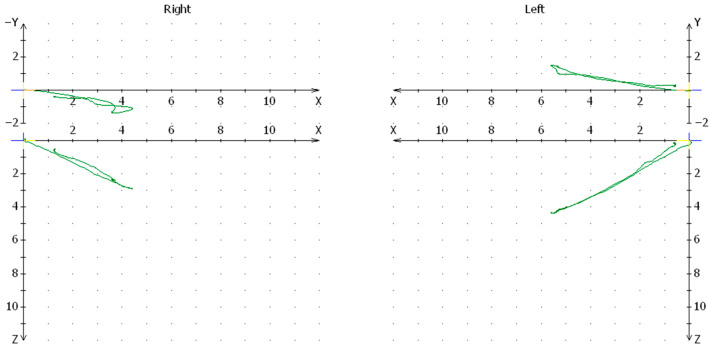
Chart of the articular path in Cadiax for one of the patients from the group treated with the communication method of head enucleation (right suction after trauma).

**Figure 5 jcm-13-03706-f005:**
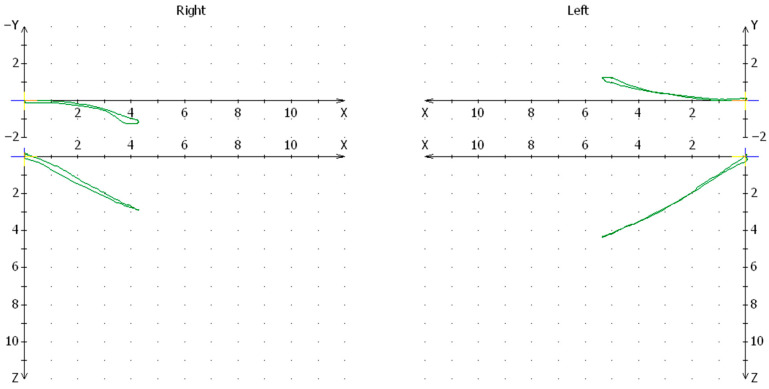
Graph of the condylar path in Cadiax of patients from the group treated with the method of removal without branch cutting (right side after trauma).

**Figure 6 jcm-13-03706-f006:**
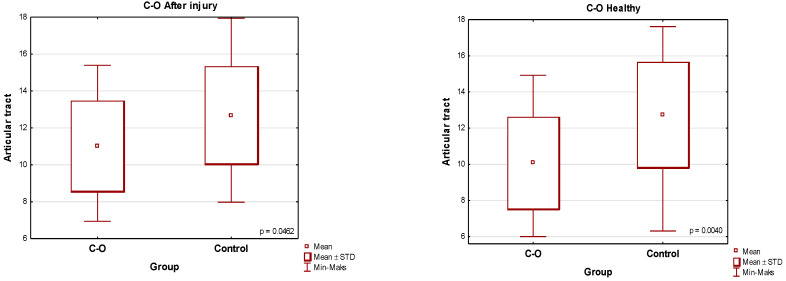
Comparison of the mean values of the articular tract [the traumatic side and the healthy side] for the conservative-orthopedic group versus the control group.

**Figure 7 jcm-13-03706-f007:**
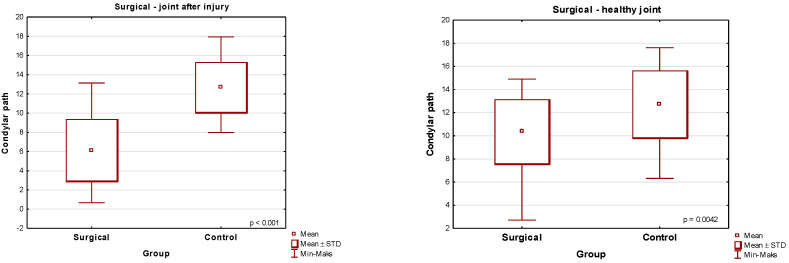
Comparison of the mean values of the condylar path on the trauma side for the surgical group versus the control group.

**Figure 8 jcm-13-03706-f008:**
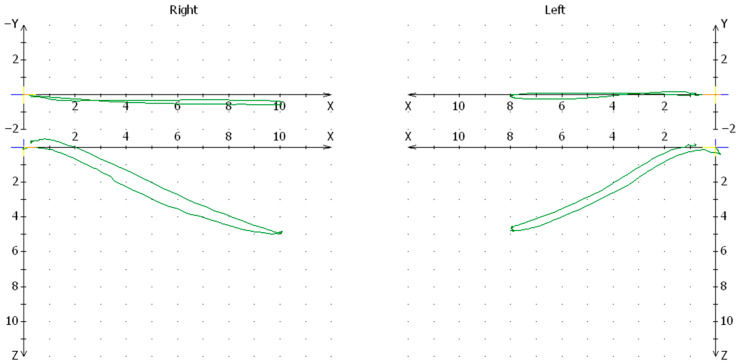
Graph of the condylar path in the Cadiax study for one of the patients from the control group.

**Table 1 jcm-13-03706-t001:** Mean values of the articular pathway in two research groups and the control group measured for the whole and healthy side, taking into account sex, age, and time from injury to treatment.

Groups			Mean Value of Condylar Path		Sex		Age	Time Injury-Treatment (Days)
			Healthy Side	Injured Side	Women	Men		
Group I—conservative treatment		20	11.01	10.07	12	8	31.70 ± 11.46	3.900 ± 2.424
Group 2—surgical treatment	Internal Fixation (IF)	12	11.54	9.19	1	11	32.92 ± 10.92	3.400 ± 2.793
	External Fixation (EF)	8	9.16	4.53	4	4	30.00 ± 14.99	5.800 ± 2.949
	Removal (R1)	13	10.94	4.71	2	11	33.54 ± 14.08	7.000 ± 6.066
	Removal with modeling (R2)	7	8.64	5.45	4	3	42.86 ± 11.16	5.000 ± 2.828
Control group		20	Healthy		12	8	26.95 ± 5.605	No treatment
			12.73	12.69				

**Table 2 jcm-13-03706-t002:** Comparison of the mean values of the articular path of the condyle in the conservative and orthopedic control group.

Conservative Treatment					
	Cases	Mean Condylar Path Value Healthy Side	Deviation	Mean Condylar Path ValueInjured Side	Deviation
Conservative	20	10.07	2.56	11.01	2.47
		Healthy Right Side		Healthy Left Side	
Control	20	12.73	2.93	12.69	2.66
Student’s *t*-test *p* < 0.001					

**Table 3 jcm-13-03706-t003:** Comparison of the general values of the articular path of the condyle in the control group and the surgical group.

Surgical Treatment					
	Cases	Mean Condylar Path Value Healthy Side	Deviation	Mean Condylar Path Value Injured Side	Deviation
Surgical	40	10.362	2804	6.152	3250
		Healthy Right Side		Healthy Left SidE	
Control	20	12.73	2.93	12.69	2.66
Student’s *t*-test *p* < 0.001					

## Data Availability

Data are contained within the article.

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
