# Peer review of "Influence of Method of Treatment of Mandibular Condylar Fractures on Range of Articular Path Measured by Cadiax Device"

_jcm, 2024, doi:10.3390/jcm13133706_

Round 1
Reviewer 1 Report
Comments and Suggestions for Authors
The study highlights the effectiveness of current treatment methods for condylar fractures, the importance of post-operative rehabilitation, and the need for further research to understand the impacts of various treatment approaches on TMJ functionality.
I have some comments:
First of all, I have to report that the authors did not adhere to the publisher's template for the manuscript.
Introduction:
The Introduction is too lengthy. Shortening it would enhance the clarity and direction of the document.
Material and Methods
A critical examination reveals a significant limitation that could impact the generalizability of its findings: the sample size. The small sample size raises questions regarding the statistical power of the study and its findings.
It becomes challenging to ensure that the sample accurately reflects the broader population of individuals suffering from condylar fractures.
Author Response
Dear Reviewer,
Thank you very much for your relevant comments and valuable suggestions. We will make every effort to comply with them.
1. we apologize, but by mistake we sent the draft version of our publication, we adjusted the guidelines; editorial template and put the work on the MPDI template
2 This includes the introduction, which is much shorter in the editorial version. The introduction has been modified and shortened
3 Indeed, a small study group may limit the reliability of statistical results. We are aware that a larger group of study patients translates into more representative data. We made a lot of limitations in selecting the selected study and control group. Patients qualified according to rigorous criteria for inclusion in the Cadiax joint pathway measurements were invited for follow-up examinations. Ultimately, the condylar pathway study was performed in 20 patients
This was a pilot study. We would like to resume the study on a larger number of participants to confirm the results.
4 Conservative as well as surgical treatment of fractured mandibular condylar processes, the move away from methods of drastic removal of fractured mandibular heads to their immobilization or replacement with standard or individual endoprostheses is highly controversial. It is a frequent topic of heated scientific debates. It often requires multispecialty treatment. Fractures of the mandible are the most common fractures that occur in the facial part of the skull. Among them, condylar fracture accounts for 20-72% of cases Treatment of condylar fractures is among the most difficult surgical procedures in craniofacial trauma, and one of the not uncommon postoperative complications noted is malocclusion. It is appropriate to say that the treatment and evaluation of mandibular condylar fractures is challenging.
5. We also added authors and fixed Affiliations. Everything is highlighted in the attachment file.
Please see the attachment.

Reviewer 2 Report
Comments and Suggestions for Authors
Thank you for the opportunity to review this article.
This is a retrospective study on the influence of the type of treatment of unilateral condylar fracture of the mandible on the mandibular movements - range of motion assessed by Cadiax device.
There are a few strengths in this article:
-
Good clinical question and relevance to current clinical practice (functional outcome following condylar fracture treatment)
-
Excellent patient selection criteria (inclusion/exclusion)
-
Objective assessment (Cadiax) and not only clinical evaluation
-
Comparison with a control group
On the other hand, there are a few issues to address:
-
This is a retrospective study (by far inferior to a prospective trial) - it should be clearly stated as a limitation.
-
Small number of cases in some of the surgical groups - problematic statistics
-
Insufficient demographic/baseline characteristics of the groups of patients - you need to prove that the outcome is not from the difference between the groups characteristics. Table 1 - There is a Polish word there - not sure what is the purpose - no statistics in table - are the groups similar or not demographically?
-
No data on whether the Cadiax assessment was done immediately or after some time from the treatment
-
Possible confounding variables that could influence the jaw motion are not discussed.
-
No reference whether surgical procedures might have changed during the 4 year period.
- Figure 2 - Legend in Polish - please fix.
- Table 2 - what is the p value?
- Figure 6+7 - Legend in Polish - please fix.
The results are not surprising but this is objective assessment and could guide the clinicians in decision making.
Overall, the article is a welcome addition to the data on a clinically relevant question. However, the retrospective study design, small numbers in some groups and insufficient methodological details decrease the level and strength of evidence and applicability of the results. Those shortcomings should be clearly stated in the article.
Author Response
Dear Reviewer,
Thank you very much for your relevant comments and valuable suggestions. We will make every effort to comply with them.
- This is a retrospective study, it is inferior to a prospective study which we are aware of. The research is still ongoing, we are continuing and improving it, however, before publishing the prospective study, we wanted to know the opinion of other researchers.
- Due to the strict criteria for the group inclusion of patients, the groups became small which may become a statistical problem, however, we want to ensure that the statistics were performed by academic researchers.
- We translated all the Polish word in Tables and Figures.
- Over the years, we have improved treatment techniques however, to standardize the study group, we qualified for this study using the same criteria
- We also added author and fixed the Affiliation
Please see the attachment.

Round 2
Reviewer 1 Report
Comments and Suggestions for Authors
The manuscript is now ready for publication.